# GigaSpeech 2: An Evolving, Large-Scale and Multi-domain ASR Corpus for Low-Resource Languages with Automated Crawling, Transcription and Refinement

## Abstract

The evolution of speech technology has been spurred by the rapid increase in dataset sizes. Traditional speech models generally depend on a large amount of labeled training data, which is scarce for low-resource languages. This paper presents GigaSpeech 2, a large-scale, multi-domain, multilingual speech recognition corpus. It is designed for low-resource languages and does not rely on paired speech and text data. GigaSpeech 2 comprises about 30,000 hours of automatically transcribed speech, including Thai, Indonesian, and Vietnamese, gathered from unlabeled YouTube videos. We also introduce an automated pipeline for data crawling, transcription, and label refinement. Specifically, this pipeline uses Whisper for initial transcription and TorchAudio for forced alignment, combined with multi-dimensional filtering for data quality assurance. A modified Noisy Student Training is developed to further refine flawed pseudo labels iteratively, thus enhancing model performance. Experimental results on our manually transcribed evaluation set and two public test sets from Common Voice and FLEURS confirm our corpus's high quality and broad applicability. Notably, ASR models trained on GigaSpeech 2 can reduce the word error rate for Thai, Indonesian, and Vietnamese on our challenging and realistic YouTube test set by 25% to 40% compared to the Whisper large-v3 model, with merely 10% model parameters. Furthermore, our ASR models trained on GigaSpeech 2 yield superior performance compared to commercial services. We believe that our newly introduced corpus and pipeline will open a new avenue for low-resource speech recognition and significantly facilitate research in this area.

## 1 Introduction

In recent years, the scaling of model parameters and data size has prevailed and proven effective in a range of areas, including language Kaplan et al. (2020); Hoffmann et al. (2022), vision Betker et al. (2023); Dehghani et al. (2023), as well as speech processing Pratap et al. (2024); Zhang et al. (2023); Radford et al. (2023). Consequently, pursuing superior AI models is now closely associated with expanding model size and leveraging larger, high-quality datasets. In the realm of Automatic Speech Recognition (ASR), several large-scale open-source labeled speech datasets Chen et al. (2021); Kang et al. (2024); Zhang et al. (2022); Galvez et al. (2021); Pratap et al. (2020b); Ardila et al. (2020) have been proposed. However, these extensive datasets are only available for several mainstream languages, such as English and Mandarin, hindering speech recognition development for low-resource languages. Moreover, traditional ASR corpus Ardila et al. (2020); Conneau et al. (2023); Bu et al. (2017); Du et al. (2018) construction relies heavily on human-labeled speech data, making it time-consuming and a major bottleneck in the fast-paced AI industry. Reducing dependence on vast labeled data is crucial when expanding to new languages and domains Hsu et al. (2021). YODAS Li et al. (2023) attempts to address this issue by building multilingual datasets via scraping audio and transcriptions from YouTube. However, neither manual nor automatic subtitles accurately reflect the speech content, resulting in unguaranteed quality.

With this perspective in mind, we propose a new paradigm for constructing large-scale ASR datasets, focusing exclusively on audio content irrespective of the existence or quality of corresponding text pairs. This approach leverages the gigantic amount of unlabeled audio data, thereby bypassing the constraints of scarce paired data. We introduce GigaSpeech 2, an evolving, large-scale, multi-domain, multilingual ASR corpus for low-resource Southeast Asian languages. *GigaSpeech 2 raw* comprises about 30,000 hours of automatically transcribed speech, including Thai, Indonesian, and Vietnamese. *GigaSpeech 2 refined* consists of 10,000 hours of Thai, 6,000 hours each for Indonesian and Vietnamese. To achieve this, an automated pipeline is developed for data crawling, transcription, and filtering. Furthermore, a modified Noisy Student Training (NST) Xie et al. (2020) method is proposed to refine labels from flawed data iteratively. Through comprehensive evaluations, ASR models trained on *GigaSpeech 2 refined* can reduce the word error rate for Thai, Indonesian, and Vietnamese on our YouTube test set by 25% to 40% compared to the powerful Whisper large-v3 model, with merely 10% model parameters.

Our contributions can be summarized as follows:

- We release GigaSpeech 2, an evolving, large-scale, multi-domain, and multilingual ASR corpus focusing on low-resource languages. *GigaSpeech 2 raw* comprises about 30,000 hours of automatically transcribed speech across Thai, Indonesian, and Vietnamese. *GigaSpeech 2 refined* consists of 10,000 hours of Thai, 6,000 hours each for Indonesian and Vietnamese.

- We develop an automated pipeline for data crawling, transcription, and label refinement, enabling the creation of large-scale speech datasets without reliance on labeled data.

- We propose a modified NST method to refine flawed pseudo labels iteratively. Our modified NST considers scaling, relabeling, and filtering data within each iteration, significantly improving data quality.

- We release a series of challenging and realistic speech recognition test sets, including Thai, Indonesian, and Vietnamese. Compared to previous public test sets, GigaSpeech 2 test sets more realistically reflect speech recognition scenarios and mirror the real performance of an ASR system for low-resource languages.

- Experimental results on our challenging GigaSpeech 2 test sets, as well as other competitive public test sets including Common Voice and FLEURS, demonstrate the superiority of the ASR models trained on GigaSpeech 2 over several competitive baselines, including Whisper large-v3 and commercial services.

## 2 RELATED WORK

**Multilingual Low-Resource Speech Datasets**   Several public multilingual speech datasets have emerged for low-resource languages. BABEL Gales et al. (2014), a pioneering dataset, includes conversational telephone data in 17 African and Asian languages. Common Voice Ardila et al. (2020) offers 19,000 hours of validated recordings in over 100 languages. FLEURS Conneau et al. (2023) covers 102 languages with 12 hours of supervised data per language. CMU Wilderness Black (2019) provides 20 hours of New Testament data for over 700 languages. VoxLingua107 Valk & Alumäe (2021) contains 6,628 hours of unlabeled YouTube data across 107 languages. However, most public multilingual speech datasets focus on high-resource languages, leaving low-resource languages with limited annotated speech data. For example, the available open-source data for Thai, Indonesian, and Vietnamese is scarce, as detailed in Table 1. In contrast, industry-utilized speech models like Whisper Radford et al. (2023), MMS Pratap et al. (2024), Google USM Zhang et al. (2023), and Universal-1 Ramirez et al. (2024) are trained on massive industrial-grade datasets, the details of which remain undisclosed. To resolve the problem, YODAS Li et al. (2023) attempts to crawl audio from YouTube, but neither manual nor automatic subtitles accurately reflect the speech content, resulting in unguaranteed quality. Moreover, widely used evaluation benchmarks for low-resource languages Ardila et al. (2020); Conneau et al. (2023) only consist of read speech, which is relatively clean and mismatched with real-world speech data.

**Multilingual Automatic Speech Recognition**   As the demand for communication between people worldwide grows, many works Radford et al. (2023); Zhang et al. (2023); Pratap et al. (2024); Li et al. (2021); Lugosch et al. (2022); Toshniwal et al. (2018); Cho et al. (2018); Pratap et al.

Table 1: Comparison of data size between GigaSpeech 2 and other common public multilingual speech datasets on Thai ("th"), Indonesian ("id"), and Vietnamese ("vi").

| Dataset | Language | Total Duration (h) | Domain | Speech Type | Labeled | Label Type |
|---|---|---|---|---|---|---|
| Common Voice Ardila et al. (2020) | th | 172.0 | Open domain | Read | Yes | Manual |
| | id | 28.0 | | | | |
| | vi | 6.0 | | | | |
| FLEURS Conneau et al. (2023) | th | 13.3 | Wikipedia | Read | Yes | Manual |
| | id | 12.6 | | | | |
| | vi | 13.3 | | | | |
| VoxLingua107 Valk & Alumäe (2021) | th | 61.0 | YouTube | Spontaneous | No | - |
| | id | 40.0 | | | | |
| | vi | 64.0 | | | | |
| CMU Wilderness Black (2019) | th | 15.6 | Religion | Read | Yes | Manual |
| | id | 70.9 | | | | |
| | vi | 9.2 | | | | |
| BABEL Gales et al. (2014) | vi | 87.1 | Conversation | Spontaneous | Yes | Manual |
| VietMed Le-Duc (2024) | vi | 16.0 | Medical | Spontaneous | Yes | Manual |
| Thai Dialect Corpus Suwanbandit et al. (2023) | th | 840.0 | Open domain | Read | Yes | Manual |
| TITML-IDN Shinoda & Furui (2011) | id | 14.5 | News | Read | Yes | Manual |
| MEDISCO Qorib & Adriani (2018) | id | 10.0 | Medical | Read | Yes | Manual |
| YODAS manual Li et al. (2023) | th | 497.1 | YouTube | Spontaneous | Yes | Manual |
| | id | 1420.1 | | | | |
| | vi | 779.9 | | | | |
| YODAS automatic Li et al. (2023) | th | 1.9 | YouTube | Spontaneous | Yes | Pseudo |
| | id | 8463.6 | | | | |
| | vi | 9203.1 | | | | |
| *GigaSpeech 2 raw* | th | 12901.8 | YouTube | Spontaneous | Yes | Pseudo |
| | id | 8112.9 | | | | |
| | vi | 7324.0 | | | | |
| *GigaSpeech 2 refined* | th | 10262.0 | YouTube | Spontaneous | Yes | Pseudo |
| | id | 5714.0 | | | | |
| | vi | 6039.0 | | | | |

(2020a); Tjandra et al. (2023); Kannan et al. (2019); Conneau et al. (2021) have shifted attention to multilingual speech recognition. Whisper Radford et al. (2023), built on 680,000 hours of web data, supports 99 languages. Google USM Zhang et al. (2023), trained on YouTube audio, extends to 100+ languages. Massively Multilingual Speech (MMS) Pratap et al. (2024), trained on religion data, further scales to 1,107 languages.

**Noisy Student Training (NST)** NST Xie et al. (2020); Park et al. (2020); Xu et al. (2020); Zhang et al. (2020); Likhomanenko et al. (2021); Mehmood et al. (2022); Chen et al. (2023) is a self-training technique that leverages unlabeled data to enhance performance. Traditional NST methods start with training a teacher model on high-quality labeled data. Each student model then trains on both noisy-augmented labeled data and pseudo-labeled data generated by its teacher from the unlabeled data. A recent study Xu et al. (2020) uses Character Error Rate (CER) between pseudo-labeled data generated with and without a language model to perform data selection, suggesting a positive correlation between the CERs of different pseudo labels and their ground truth.

## 3 DATASET CONSTRUCTION

Our proposed automated construction pipeline is illustrated in Fig. 1. Sec. 3.1 covers the stages involved in building *GigaSpeech 2 raw* and Sec. 3.2 further construct *GigaSpeech 2 refined*.

### 3.1 GIGASPEECH 2 RAW: AUTOMATED CRAWLING AND TRANSCRIPTION

**Audio Collection** Due to the scarcity of human-labeled data in low-resource languages, our dataset is collected with a focus solely on the audio content, irrespective of the existence or quality of corresponding text pairs. This strategy allows for a broader range of audio data. Given the scarcity

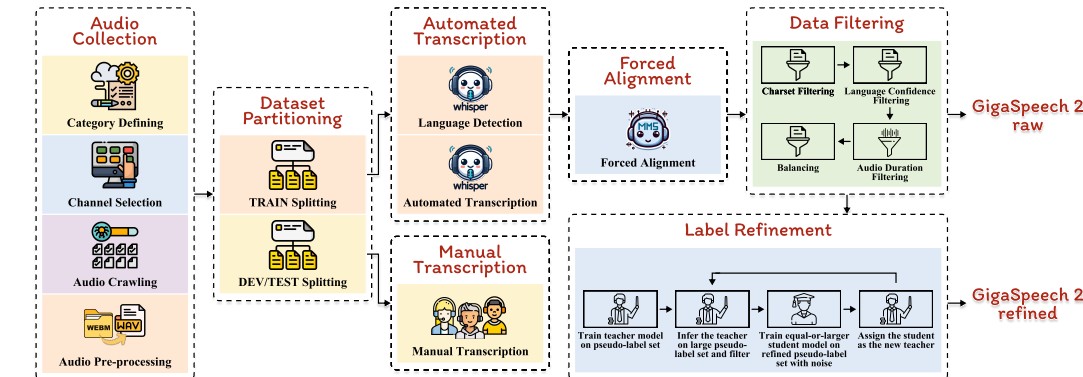

Figure 1: Automated construction pipeline of GigaSpeech 2, comprising (1) audio collection, (2) dataset partitioning, (3) automated transcription with Whisper, (4) forced alignment with TorchAudio, (5) transcription normalization, (6) data filtering, and (7) label refinement.

and uneven distribution of resources for low-resource languages, we strategically focus on crawling videos from YouTube channels based on two key assumptions. First, prioritizing popular channels ensures consistent domain characteristics and audio quality. Second, different channels have no speaker overlap, simplifying the subsequent data partitioning. The data collection process starts by manually defining categories of interest. The selected topics include Agriculture, Art, Business, Climate, Culture, Economics, Education, Entertainment, Health, History, Literature, Music, Politics, Relationships, Shopping, Society, Sport, Technology, and Travel. Alongside multiple topics, various content formats are also considered, including Audiobook, Commentary, Lecture, Monologue, Movie, News, Talk, and Vlog. This broad selection ensures the comprehensiveness of the dataset across multiple domains for research and analysis. Moreover, **the collected audio must be accompanied by a Creative Commons license**. Once the list of YouTube channels is prepared, we use yt-dlp[1] toolkit to download all audio files in WebM format. These files are then converted to WAV format with a single channel and resampled at a 16 kHz sampling rate.

**Creating TRAIN/DEV/TEST Splits**   To ensure no speaker overlap between the splits, we manually verify no speaker overlap between different channels and partition the data by allocating different YouTube channels to each subset. The dataset is divided into three distinct subsets: TRAIN, DEV, and TEST. The DEV and TEST sets each contain 10 hours and are manually transcribed by professionals, while the remainder is allocated to the TRAIN set. Table 1 shows the amount of data across these three languages. Detailed analysis of GigaSpeech 2 is illustrated in Appendix B.

**Transcription with Whisper**   Whisper large-v3 model[2] from OpenAI is used to transcribe audio files automatically. For each audio recording, a 30-second segment is selected from the middle to perform language detection by Whisper. Only audios that match the target languages are transcribed.

**Forced Alignment with TorchAudio**   Although Whisper can generate timestamps, inspection reveals they are not precise enough. We resort to the model[3] from TorchAudio Hwang et al. (2023) for forced alignment, which provides reliable alignment for noisy transcriptions, supports efficient processing on GPUs, and handles longer sequences more effectively (Pratap et al., 2024).

**Text Normalization**   Text normalization on transcripts involves applying Normalization Form Compatibility Composition (NFKC), converting all characters to uppercase, removing punctuation, and mapping Arabic numerals to corresponding words in the respective languages.

**Multi-dimensional Filtering**   A series of heuristic filtering rules across text and audio modalities are implemented to exclude relatively poor-quality samples.

- **Charset Filtering:** Segments are retained if they only contain characters permitted by the charset of the respective language.

---

[1] https://github.com/yt-dlp/yt-dlp
[2] https://huggingface.co/openai/whisper-large-v3
[3] https://dl.fbaipublicfiles.com/mms/torchaudio/ctc_alignment_mling_uroman/model.pt

- **Language Confidence Filtering:** The language identification (LID) model[4] from fast-Text (Joulin et al., 2016) is used to filter based on the estimated language confidence score, retaining only segments with confidence scores above a predetermined threshold. This method effectively eliminates meaningless and repetitive segments. Note that language identification based on audio has already been performed before transcription.

- **Audio Duration Filtering:** Segments are filtered based on duration, with only those retained within the predetermined minimum and maximum duration thresholds.

- **Balancing:** We carefully control the duplication of transcripts caused by channel-specific content while preserving natural linguistic patterns. Samples containing personal information, such as phone numbers, ID numbers, and specific addresses, are removed.

### 3.2 GigaSpeech 2 refined: Iterative Label Refinement

Some samples remain low quality due to inaccuracies in Whisper transcriptions and imprecise forced alignment boundaries. To address this, we develop a modified NST method. As illustrated in the bottom right corner of Fig. 1, it begins by training a teacher model on a subset of flawed pseudo labels, iteratively expanding the training set, generating new pseudo labels, and filtering them. A student model, equal to or larger than the teacher, is trained on these refined pseudo labels and assigned as the new teacher. Unlike previous NST approaches that heavily rely on unchanged supervised data combined with additional unsupervised data, our method eliminates the need for supervised data as a seed. Instead, we treat the flawed pseudo labels generated by Whisper as supervised data, refining all labels iteratively based on the Character Error Rate (CER) between those produced by Whisper and the teacher model. SpecAugment (Park et al., 2019), Bypass (Yao et al., 2024), and feature mask (Yao et al., 2024) introduce noise during each NST step. Bypass, a type of stochastic depth, learns channel-wise scalar weights to combine the module input and output. Feature mask performs dropout in the hidden dimension of the feedforward and convolution layer but shares across the time dimension. This deliberate noising enables the student model to learn consistency with the teacher model, which remains unaffected by noise when generating pseudo labels (Xie et al., 2020). This iterative process progressively enhances data quality. Detailed algorithm steps are provided in Appendix A Algo. 1.

## 4 Experiments

### 4.1 ASR Model Training on GigaSpeech 2

Our ASR systems are constructed by Zipformer Transducer Graves et al. (2013). Two Zipformer Yao et al. (2024) variants, namely Zipformer-M and Zipformer-L, are employed for each NST iteration. Specific configurations are listed in Appendix C.1. During Noisy Student Training, SpecAugment Park et al. (2019) is used as input noise, and Bypass Yao et al. (2024) and feature mask Yao et al. (2024) are used as model noise.

Table 2 presents the ASR results across different NST iterations on three evaluation sets, including the development and test sets from GigaSpeech 2 and the Common Voice 17.0 and FLEURS test set. Each iteration involves distinct modifications aimed at refining high-quality transcriptions. A subset of automatic transcriptions generated by Whisper large-v3 is used to train the initial teacher model (Iteration 1). The teacher model then filters the training utterances by applying a CER/WER threshold, using the original labels as references and the new labels generated by the teacher as the hypothesis. The student model is trained on this filtered set with noise injected (Iteration 2). The student model is then used as the teacher to generate new labels on a larger subset of raw automatic transcriptions, applying the same filter to refine the training data. This refined data is used to train the student model with noise injected (Iteration 3). The process repeats in subsequent iterations, and the model size is scaled up to a larger version in the final iteration (Iteration 3 of Indonesian & Vietnamese, Iteration 4 of Thai).

According to the results shown in Table 2, several notable trends can be observed:

---

[4]https://dl.fbaipublicfiles.com/fasttext/supervised-models/lid.176.bin

Table 2: Comparison of ASR performance with different NST iterations on various evaluation sets, including GigaSpeech 2 DEV and TEST, Common Voice 17.0 TEST, and FLEURS TEST. Detailing training set size (# Hours), model size (# Params), Character Error Rate (CER) for Thai, and Word Error Rate (WER) for Indonesian and Vietnamese.

| NST Iter | # Hours (h) | # Vocab | # Params (M) | CER / WER | | | |
| --- | --- | --- | --- | --- | --- | --- | --- |
| | | | | GigaSpeech 2 | | Common Voice | FLEURS |
| | | | | DEV | TEST | TEST | TEST |
| **Thai** | | | | | | | |
| 1 | 4378 | 500 | 65.5 | 12.14 | 15.10 | 8.88 | 14.33 |
| 2 | 3497 | 500 | 65.5 | $10.97_{-9.6\%}$ | $13.15_{-12.9\%}$ | $6.99_{-21.3\%}$ | $11.93_{-16.7\%}$ |
| 3 | 7219 | 2000 | 68.6 | $10.50_{-4.3\%}$ | $12.46_{-5.2\%}$ | $4.61_{-34.0\%}$ | $10.94_{-8.3\%}$ |
| 4 | 10262 | 2000 | 151.9 | $10.45_{-0.5\%}$ | $12.46_{-0.0\%}$ | $4.15_{-10.0\%}$ | $10.54_{-3.7\%}$ |
| **Indonesian** | | | | | | | |
| 1 | 5765 | 2000 | 68.6 | 16.68 | 15.99 | 19.82 | 16.29 |
| 2 | 4534 | 2000 | 68.6 | $15.60_{-6.5\%}$ | $15.23_{-4.8\%}$ | $15.83_{-20.1\%}$ | $14.30_{-12.2\%}$ |
| 3 | 5714 | 2000 | 151.9 | $14.58_{-6.5\%}$ | $14.92_{-2.0\%}$ | $13.83_{-12.6\%}$ | $13.77_{-3.7\%}$ |
| **Vietnamese** | | | | | | | |
| 1 | 2351 | 2000 | 68.6 | 16.08 | 16.95 | 24.63 | 17.86 |
| 2 | 1764 | 2000 | 68.6 | $15.08_{-6.2\%}$ | $14.72_{-13.2\%}$ | $18.81_{-23.6\%}$ | $13.50_{-24.4\%}$ |
| 3 | 6039 | 2000 | 151.9 | $14.09_{-6.6\%}$ | $12.83_{-12.8\%}$ | $14.43_{-23.3\%}$ | $11.59_{-14.1\%}$ |

1) Across all three languages (Thai, Indonesian, and Vietnamese), iteratively scaling the training data size, adding noise, and filtering labels lead to consistent improvements in the WER performance on the evaluation sets until the final iteration. This indicates that the iterative approach of refining and scaling the training data is effective in enhancing the accuracy of the raw transcriptions.

2) The Thai language achieves the absolute lowest error rates consistently across iterations from Iteration 1 to 4, indicating the effectiveness of the NST approach for this particular language. The best NST model outperforms the standard transcription model data by WER reductions of 1.69%, 2.64%, 4.73%, and 3.79% absolute (13.92%, 17.48%, 53.27%, and 26.45% relative) respectively (Iteration 4 *vs.* 1).

Additional ablation studies on our modified NST in Appendix D Table 9 demonstrate the effectiveness of relabeling and discuss the detriment of enlarging noise when scaling the training data.

## 4.2 Comparison to Existing ASR Systems

To demonstrate the efficacy of our ASR models trained on GigaSpeech 2, several mainstream and competitive ASR systems, including Whisper Radford et al. (2023) from OpenAI, MMS Pratap et al. (2024) from Meta, and commercial services from Azure and Google, are used as benchmarks.

**Whisper:** Our work builds upon Whisper Radford et al. (2023), a suite of large-scale, multitask, and multilingual speech models developed by OpenAI. It leverages the encoder-decoder Transformer architecture Vaswani et al. (2017), with model sizes ranging from 39 million parameters (tiny) to 1.55 billion parameters (large). Additionally, Whisper offers variants spanning from an English-only version to a multilingual model capable of handling 99 languages. To conduct a comprehensive evaluation, we test three variants: Whisper base, Whisper large-v2, and Whisper large-v3 models.

**MMS:** The Massively Multilingual Speech (MMS) Pratap et al. (2024) project leverages self-supervised learning (SSL) techniques and a novel dataset to expand the language coverage of speech technology significantly. The core components include pre-trained wav2vec 2.0 Baevski et al. (2020) models for 1,406 languages, a single multilingual ASR model supporting 1,107 languages, speech synthesis models for the same set of languages, and a language identification model capable of recognizing 4,017 languages. In this study, we employ the MMS L1107 configuration.

**Azure AI Speech:** Azure Speech CLI offers a convenient way to leverage Microsoft's speech recognition capabilities directly from the command line. It not only supports a wide range of audio file formats but also possesses the ability to handle various streaming audio inputs. We utilize the Azure Speech CLI version 1.37 in this paper, which is the latest version available.

Table 3: Comparison of ASR results for models trained on GigaSpeech 2 with open-source multilingual ASR models and commercial ASR services, evaluated on test sets from GigaSpeech 2, Common Voice 17.0, and FLEURS. The evaluation metrics are Character Error Rate (CER) for Thai and Word Error Rate (WER) for both Indonesian and Vietnamese. "†" denotes commercial services.

| Model | # Params (M) | CER / WER | | |
| --- | --- | --- | --- | --- |
| | | GigaSpeech 2 | Common Voice | FLEURS |
| **Thai** | | | | |
| Whisper large-v3 | 1542 | 20.44 | 6.02 | 11.55 |
| Whisper large-v2 | 1541 | 22.47 | 8.79 | 15.50 |
| Whisper base | 72 | 46.47 | 32.59 | 42.28 |
| MMS L1107 | 964 | 31.75 | 14.49 | 23.07 |
| Azure Speech CLI 1.37.0† | - | 17.25 | 10.20 | 13.35 |
| Google USM Chirp v2† | - | 49.70 | 14.75 | 63.35 |
| GigaSpeech 2 (proposed) | 151.9 | **12.46** | **4.15** | **10.54** |
| **Indonesian** | | | | |
| Whisper large-v3 | 1542 | 20.03 | 7.43 | 7.85 |
| Whisper large-v2 | 1541 | 21.44 | 8.93 | 8.95 |
| Whisper base | 72 | 39.37 | 34.70 | 33.76 |
| MMS L1107 | 964 | 35.27 | 20.72 | 24.49 |
| Azure Speech CLI 1.37.0† | - | 18.07 | 10.33 | 11.18 |
| Google USM Chirp v2† | - | 19.63 | 9.70 | **7.23** |
| GigaSpeech 2 (proposed) | 151.9 | **14.92** | 13.83 | 13.77 |
| + Common Voice + FLEURS | 151.9 | 14.95 | **7.33** | 12.74 |
| **Vietnamese** | | | | |
| Whisper large-v3 | 1542 | 17.94 | 13.74 | **8.59** |
| Whisper large-v2 | 1541 | 18.74 | 18.00 | 10.26 |
| Whisper base | 72 | 39.88 | 44.07 | 40.41 |
| MMS L1107 | 964 | 46.62 | 43.88 | 55.35 |
| Azure Speech CLI 1.37.0† | - | **11.86** | **10.21** | 11.88 |
| Google USM Chirp v2† | - | 13.28 | 12.46 | 11.75 |
| GigaSpeech 2 (proposed) | 151.9 | 12.83 | 14.43 | 11.59 |
| + Common Voice + FLEURS | 151.9 | 12.39 | 11.47 | 9.94 |

**Google USM:** The Universal Speech Model (USM) Zhang et al. (2023) is introduced as a single, large-scale model that excels in ASR across over 100 languages. This achievement is made possible by pre-training the model's encoder on a vast, unlabeled multilingual dataset of 12 million hours, covering more than 300 languages, followed by fine-tuning on a smaller labeled dataset. To conduct a thorough comparison, we utilize their Chirp Speech-to-Text v2 model for performance evaluation.

We compare the performance of our proposed approach trained on GigaSpeech 2 against these above-mentioned ASR models, including Whisper (base, large-v2, and large-v3), MMS L1107, Azure Speech CLI 1.37.0 and Google USM Chirp v2[5], across three languages: Thai, Indonesian, and Vietnamese. The ASR performance is evaluated regarding character error rate (CER) or word error rate (WER) on three distinct test sets from GigaSpeech 2, Common Voice 17.0, and FLEURS. According to the results shown in Table 3, there are several intriguing findings:

1) For the Thai language, our ASR model trained on GigaSpeech 2 (Table 3, Thai, Row 7) outperforms all competitors, including commercial services from Azure and Google, securing the top rank across all three test sets among the seven models. It outperforms Whisper large-v3 by WER reductions of 7.98%, 1.87%, and 1.01% absolute (39.04%, 31.06%, and 8.74% relative) (Table 3, Thai, Row 7 *vs.* 1). Remarkably, our model achieves such impressive performance with nearly one-tenth of the parameters compared to Whisper large-v3 (151.9 M *vs.* 1542 M).

---

[5]Abnormal high deletion rates with Google USM in Thai are observed in our repeated testing.

Table 4: Comparison of ASR results for models trained on YODAS and GigaSpeech 2, evaluated on test sets from GigaSpeech 2, Common Voice 17.0, and FLEURS. The evaluation metrics are Character Error Rate (CER) for Thai and Word Error Rate (WER) for both Indonesian and Vietnamese.

| Training Set | # Params (M) | CER / WER | | |
| --- | --- | --- | --- | --- |
| | | GigaSpeech 2 | Common Voice | FLEURS |
| **Thai** | | | | |
| YODAS manual | 68.6 | 27.34 | 10.71 | 14.19 |
| YODAS manual | 151.9 | 28.76 | 10.96 | 16.11 |
| *GigaSpeech 2 refined* | 151.9 | **12.46** | **4.15** | **10.54** |
| **Indonesian** | | | | |
| YODAS manual | 68.6 | 25.77 | **10.82** | 14.63 |
| YODAS manual + automatic | 68.8 | 41.11 | 15.41 | 47.26 |
| YODAS manual | 151.9 | 25.11 | 11.05 | **12.67** |
| *GigaSpeech 2 refined* | 151.9 | **14.92** | 13.83 | 13.77 |
| **Vietnamese** | | | | |
| YODAS manual | 68.6 | 40.35 | 31.07 | 25.68 |
| YODAS manual + automatic | 68.6 | 71.91 | 25.73 | 61.38 |
| YODAS manual | 151.9 | 40.71 | 32.58 | 29.32 |
| *GigaSpeech 2 refined* | 151.9 | **12.83** | **14.43** | **11.59** |

2) For the Indonesian and Vietnamese languages, our system demonstrates competitive performance compared to existing baseline models. This highlights the efficacy of our pipeline in delivering high-quality results with a lightweight model. Specifically, on the GigaSpeech 2 test set in the Indonesian language, our system (Table 3, Indonesian, Row 7) outperforms all baseline models, attaining the best performance. Compared to Whisper large-v3, the model trained on Indonesian achieves an absolute WER reduction of 5.11%, corresponding to a relative reduction of 25.51% (Table 3, Indonesian, Row 7 *vs.* 1). Similarly, the model trained on Vietnamese achieves an absolute WER reduction of 5.11%, corresponding to a relative reduction of 28.48% (Table 3, Vietnamese, Row 7 *vs.* 1).

3) Our model exhibits degraded performance compared to commercial ASR systems on the Common Voice and FLEURS test sets in Indonesian and Vietnamese, which can be attributed to the domain mismatch. Contrastively, we observe a performance leap after adding Common Voice and FLEURS training data into GigaSpeech 2 (Table 3, Indonesian & Vietnamese, Row 7 *vs.* 8).

Despite the substantial disparity in training data size, our method achieves the best performance for the Thai language domain and delivers comparable results to commercial models for Indonesian and Vietnamese. This remarkable accomplishment highlights the efficacy of our approach in leveraging limited, free, open-source, unlabeled data to train highly competitive speech recognition models. It showcases a promising path towards developing high-quality speech recognition systems without the need for extensive, proprietary datasets, thereby reducing the barrier to entry and enabling wider accessibility.

## 4.3 COMPARISON TO THE YODAS CORPUS

Table 4 compares ASR performance across different models trained on YODAS Li et al. (2023) and GigaSpeech 2 datasets evaluated on various test sets. Note that YODAS Thai automatic is not included because of insufficient data (only 1 hour). Despite variations in overall data volume, several general conclusions can be drawn from the trend analysis:

1) The models trained on *GigaSpeech 2 refined* yield generally superior results compared to those trained on the YODAS datasets for all three languages.

2) The YODAS manual may suffer from overfitting or noisy data issues due to simplistic filtering rules, leading to inconsistent performance in Indonesian (Table 4, Indonesian, Row 1 & 3).

Table 5: Comparison of ASR models trained on GigaSpeech 2 with Icefall and ESPnet toolkits, evaluated on GigaSpeech 2 TEST set. The evaluation metrics are Character Error Rate (CER) for Thai (th) and Word Error Rate (WER) for both Indonesian (id) and Vietnamese (vi).

| Toolkit | Model | # Params (M) | CER / WER | | |
|---------|-------|--------------|-----------|-----|-----|
| | | | th | id | vi |
| Icefall | Zipformer/Stateless Pruned RNN-T | 151.9 | 12.46 | 14.92 | 12.83 |
| ESPnet | Conformer/Transformer CTC/AED | 111.8 | 13.70 | 15.50 | 14.60 |

3) Purely automatic generation of YODAS tends to degrade performance, as observed for Vietnamese (Table 4, Vietnamese, Row 1 *vs.* 2) and Indonesian (Table 4, Indonesian, Row 1 *vs.* 2), likely due to the inherent noise and errors in the automatically generated subtitles.

### 4.4 TRAINING ASR MODELS WITHIN ESPNET AND ICEFALL ON GIGASPEECH 2

**Icefall:** The neural Transducer Graves et al. (2013) architecture is employed, with Zipformer-L as the encoder and the pruned RNN-T loss Kuang et al. (2022) as the object function. 2000-class Byte Pair Encoding (BPE) Sennrich et al. (2016) word pieces are used. More details are provided in Appendix C.1.

**ESPnet:** The Conformer Gulati et al. (2020) CTC/AED Kim et al. (2017) system is adopted from ESPnet Watanabe et al. (2018), with Conformer-L as the encoder and a combination of the localized sensitivity of convolutional neural networks and the long-range modeling capabilities of Transformers Vaswani et al. (2017). 2000-class BPE word pieces are used. More details can be found in Appendix C.2.

Table 5 shows the results of ASR models trained with icefall and ESPnet. The models trained with ESPnet are slightly worse than icefall in all three languages, which is as expected and can be explained by the discrepancy in the number of model parameters (112M *vs.* 152M). It is worth noting that the results in Table 5 are intended to provide baseline systems for these two popular toolkits to demonstrate the universality of GigaSpeech 2 instead of pursuing state-of-the-art performance.

## 5 LIMITATION AND FUTURE WORK

Due to time constraints, we only tested 3-4 iterations of the proposed NST model. We are optimistic that more iterations will yield even better results. We are actively extending our language coverage by incorporating additional languages, including Malay, Korean, Arabic, Cantonese, and Minnan. We will also expand our low-resource language family in our future investigation. In addition, we did not perform language model fusion to further boost performance since there is a lack of high-quality and in-domain text data for low-resource languages. To resolve potential legal risks, our dataset adopts the same terms as GigaSpeech Chen et al. (2021), restricting use to non-commercial research and educational purposes only.

## 6 CONCLUSION

This paper introduces a new multilingual speech dataset, GigaSpeech 2, and a novel automated pipeline to boost speech recognition performance using in-the-wild audio-only data. GigaSpeech 2 aims to address the scarcity of labeled training data on low-resource languages by developing this large-scale, multi-domain, and multilingual corpus. Extensive experiments are conducted to validate the efficacy of our newly introduced corpus. The ASR models trained in three languages, which are Thai, Indonesian, and Vietnamese within GigaSpeech 2, demonstrate superior and impressive performance compared to various powerful ASR models, including Whisper large v2/v3 from OpenAI, MMS from Meta, and even commercial services from Google and Azure. The related resources, including the training corpus, curated test sets, automated pipeline, and recipes, will be released to facilitate research in this direction. In the future, we are eager to extend our paradigm to more low-resource languages and are devoted to breaking down the language barrier.

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

# A  ALGORITHM OF ITERATIVE LABEL REFINEMENT

Algo. 1 illustrates the workflow of our proposed iterative label refinement.

---

**Algorithm 1:** Iterative Label Refinement

---

**Input:** Pseudo-label set $\mathcal{P}$, Number of iterations $n$, Threshold $\tau$
**Output:** Refined-label set $\mathcal{R}$
Divide $\mathcal{P}$ into $n$ splits $\mathcal{P}_1, \mathcal{P}_2, \ldots, \mathcal{P}_n$;
$\mathcal{R} \leftarrow \mathcal{P}_1$;
Train teacher model $\mathcal{M}_1$ on $\mathcal{R}$ with noise;
**for** $i \leftarrow 1$ **to** $n$ **do**
  $\mathcal{R} \leftarrow \varnothing$;
  **if** $i == 1$ **then**
    // Filter $\mathcal{P}_i$ by teacher model $\mathcal{M}_i$ with CER $\leq \tau$
    $\mathcal{R} \leftarrow \{(x, y) \in \mathcal{P}_i \mid \mathrm{CER}(y, \mathcal{M}_i(x)) \leq \tau\}$;
  **else**
    **for** $j \leftarrow 1$ **to** $i$ **do**
      // Relabel $\mathcal{P}_j$ by teacher model $\mathcal{M}_i$ and filter with CER $\leq \tau$
      $\mathcal{R}_{tmp} \leftarrow \{(x, \mathcal{M}_i(x)) \mid (x, y) \in \mathcal{P}_j, \mathrm{CER}(y, \mathcal{M}_i(x)) \leq \tau\}$;
      $\mathcal{R} \leftarrow \mathcal{R} \cup \mathcal{R}_{tmp}$;
    **end**
  **end**
  Train equal-or-larger student model $\mathcal{M}_{i+1}$ on $\mathcal{R}$ with noise and assign as new teacher;
**end**
**return** $\mathcal{R}$;

---

# B  DETAILED ANALYSIS OF GIGASPEECH 2

## B.1  MANUAL TRANSCRIPTION QUALITY ASSURANCE

The manual transcription process, carried out by a professional data annotation company, includes rigorous manual quality checks and secondary inspections to ensure that timestamp accuracy and transcription correctness exceed 97%. All manually transcribed results undergo a 100% manual quality inspection, where both timestamps and transcription accuracy are thoroughly checked. Any data that fails to meet the required standards is sent back for correction. Subsequently, 30% of each inspector's reviewed data is re-evaluated. If this recheck confirms over 97% accuracy, the data passes; otherwise, the entire dataset inspected by that quality inspector is returned for full correction. For timestamp accuracy, an audio snippet tool is used to ensure that timestamps do not overlap with the waveform. If any timestamp does fall on the waveform, a manual inspection is conducted to confirm whether it corresponds to speech.

## B.2  DOMAIN DISTRIBUTION OF MANUAL EVALUATION SETS

The domain distribution of the manual evaluation sets is shown in Fig. 2. The domains are identified based on a predefined set of categories. Each sample is manually annotated at the individual video level, considering both the topic type and content format.

## B.3  DURATION DISTRIBUTION OF TRAINING SETS

The utterance-level duration distribution of the training sets is illustrated in Fig. 3.

## B.4  EVALUATION OF PROCESSING TIME

The processing times for transcription, forced alignment, filtering, segmentation, and relabeling are measured on an idle single V100 32G GPU machine using a 100-hour subset of Thai audio. The processing time and the real-time factor (RTF) are detailed in Table 6.

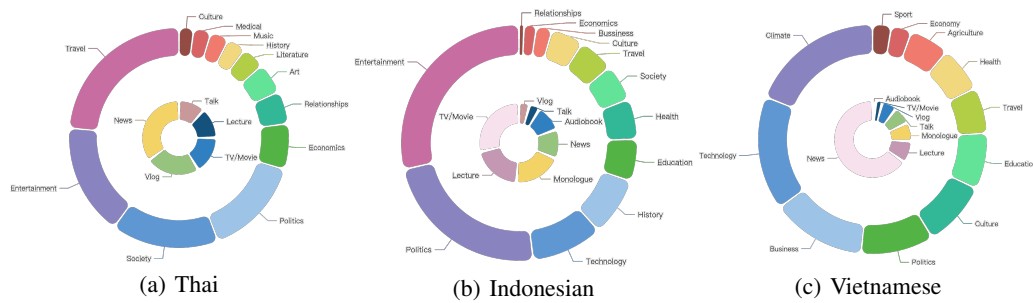

(a) Thai      (b) Indonesian      (c) Vietnamese

Figure 2: Hours distribution of manual evaluation sets for Thai, Indonesian, and Vietnamese. The inner circle represents the format, and the outer circle represents the topic.

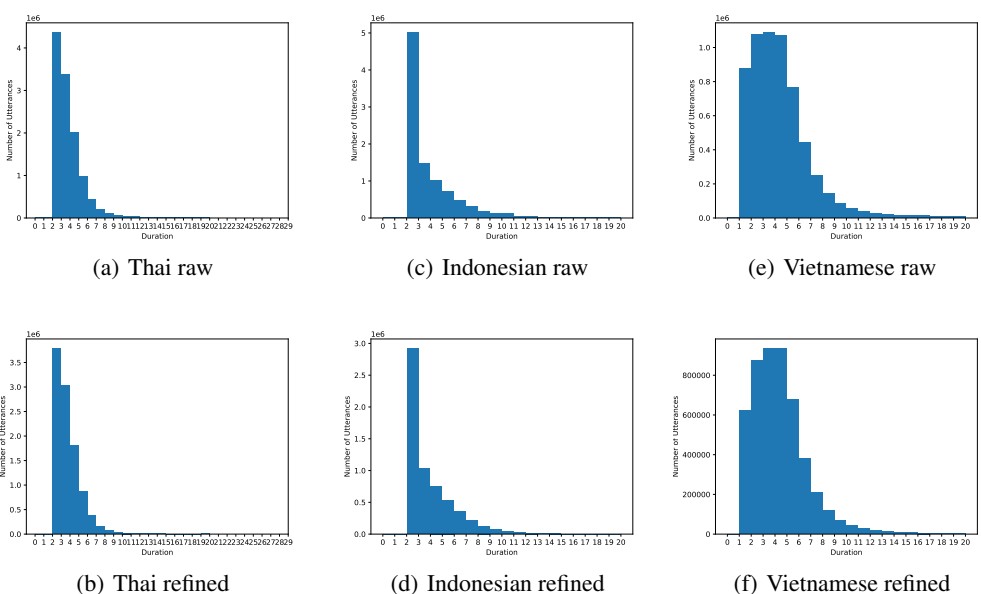

(a) Thai raw      (c) Indonesian raw      (e) Vietnamese raw

(b) Thai refined      (d) Indonesian refined      (f) Vietnamese refined

Figure 3: Utterance-level duration (second) distribution of training sets for Thai, Indonesian, and Vietnamese.

## C  MODEL CONFIGURATIONS

### C.1  CONFIGURATION OF ZIPFORMER

Two Zipformer-based models are used, following official configurations reported in icefall[6]. In each Zipformer stack, the hidden dimensions of the first and last feedforward modules are 3/4 and 5/4 of the middle one, respectively. Ahead of the encoder, a convolution subsampling module with a stride of 2 reduces the frame rate to 50 Hz. The input consists of 80-channel FBank features extracted over windows of 25ms, strided by 10ms. The label decoder utilizes a stateless decoder Ghodsi et al. (2020). 8 V100 32G GPUs are used for training. Detailed configurations are provided in Table 7.

### C.2  CONFIGURATION OF CONFORMER

A Conformer-based model is developed adhering to the official configurations outlined in ESP-net[7]. The model comprises an encoder that employs the Conformer architecture and a decoder that leverages the Transformer architecture. Moreover, the parameters for both the encoder and decoder

---

[6] https://github.com/k2-fsa/icefall
[7] https://github.com/ESPnet/ESPnet

Table 6: Evaluation of overall processing time and real-time factor (RTF) for each process in the construction of GigaSpeech 2. The processing times for transcription, forced alignment, filtering, segmentation, and relabeling are measured on an idle single V100 32G GPU machine using a 100-hour subset of Thai audio.

| Process | Time Consumption | RTF |
|---|---|---|
| Transcription | 19h 42min 13s | $1.97 \times 10^{-1}$ |
| Forced Alignment | 3h 27min 29s | $3.46 \times 10^{-2}$ |
| Filter | 3s | $8.00 \times 10^{-6}$ |
| Segmentation | 6min 58s | $1.16 \times 10^{-3}$ |
| Relabel | 40min 48s | $6.80 \times 10^{-3}$ |

components, the optimization process, the scheduling mechanism, and SpecAugment settings are carefully designed to ensure a comprehensive and efficient model setup. 4 A100 80G GPUs are used for training. The specifics of these configurations are detailed in Table 8.

## D  ABLATION STUDY ON NOISY STUDENT TRAINING

Based on the ablation study of our proposed NST on the evaluation sets in Table 9, we can analyze the effects of different iterations and their impact on performance: 1) Relabeling the data during the transition from iteration 2 to 3 is crucial for improving performance (Sys.1 *vs.* Sys.2). 2) Larger augmentation applied in our NST process may have a negative impact on the performance (Sys.1 *vs.* Sys.3). These findings suggest that careful consideration of the relabeling and augmentation strategies is crucial for optimizing the performance of the NST model across different evaluation sets and domains.

Table 7: Configuration of Zipformer at two different scales

|  | Zipformer-M | Zipformer-L |
|---|:---:|:---:|
| **Encoder** | | |
| number of stacks | 6 | |
| numbers of layers | 2,2,3,4,3,2 | 2,2,4,5,4,2 |
| downsampling factors | 1,2,4,8,4,2 | |
| output downsampling factor | 2 | |
| embedding dimensions | 192,256,384,512,384,256 | 192,256,512,768,512,256 |
| embedding unmasked dimensions | 192,192,256,256,256,192 | 192,192,256,320,256,192 |
| feedforward dimensions | 512,768,1024,1536,1024,768 | 512,768,1536,2048,1536,768 |
| convolution kernel sizes | 31,31,15,15,15,31 | |
| attention heads | 4,4,4,8,4,4 | |
| attention query dimension | 32 | |
| attention value dimension | 12 | |
| positional encoding embedding dimension | 48 | |
| projected positional encoding dimension per head | 4 | |
| **Decoder** | | |
| embedding dimensions | 512 | |
| context size | 2 | |
| **Joiner** | | |
| embedding dimensions | 512 | |
| **Criterion** | | |
| use ctc head | false | |
| use transducer head | true | |
| pruned range | 5 | |
| loss smoothing lm scale | 0.25 | |
| loss smoothing am scale | 0.0 | |
| simple loss scale | 0.5 | |
| simple loss scale warmup steps | 2000 | |
| **Frontend** | | |
| n fft | 512 | |
| hop length | 256 | |
| feature dimension | 80 | |
| **Training** | | |
| use amp | true | |
| max epochs | 30 | |
| max duration per batch | 1000 | |
| ref duration | 600 | |
| seed | 42 | |
| **Optimization** | | |
| optimizer | scaledadam | |
| base learning rate | 0.045 | |
| seed | 42 | |
| **Scheduler** | | |
| scheduler | eden | |
| lr batches | 7500 | |
| lr epochs | 10000 / training set hours | |
| warmup batches | 500 | |
| warmup starting lr | 0.5 | |
| **SpecAugment** | | |
| time warping factor | 80 | |
| number of time masks | 10 | |
| time mask maximum width | 100 | |
| number of frequency masks | 2 | |
| frequency mask width range | 0 - 27 | |

Table 8: Configuration of Conformer at the large scale.

| Conformer-L | | | |
|---|---|---|---|
| **Encoder** | | **Criterion** | |
| attention head | 8 | ctc weight | 0.3 |
| numbers of blocks | 12 | label smoothing | 0.1 |
| linear unit | 2048 | length normalized | false |
| dropout rate | 0.1 | **Frontend** | |
| positional dropout rate | 0.1 | n fft | 512 |
| attention dropout rate | 0.1 | hop length | 256 |
| input layer | conv2d | **Training** | |
| normalize before | true | use amp | true |
| macaron style | true | gradient accumulation | 4 |
| relative position type | latest | max epochs | 20 |
| position encoding layer | rel_pos | **Optimization** | |
| self-attention layer | rel_selfattn | optimizer | adam |
| activation type | swish | learning rate | 0.0025 |
| use cnn module | true | weight decay | 0.000001 |
| cnn module kernel | 31 | **Scheduler** | |
| **Decoder** | | scheduler | warmuplr |
| attention heads | 8 | warmup steps | 40000 |
| linear units | 2048 | **SpecAugment** | |
| number of blocks | 6 | time warp window | 5 |
| dropout rate | 0.1 | frequency mask width range | 0 - 27 |
| positional dropout rate | 0.1 | number of frequency masks | 2 |
| self-attention dropout rate | 0.1 | time mask width ratio range | 0.0 - 0.05 |
| source attention dropout rate | 0.1 | number of time masks | 10 |

Table 9: Ablation study of NST on GigaSpeech 2 Thai, evaluated across various evaluation sets: GigaSpeech 2 DEV and TEST, Common Voice 17.0 TEST, and FLEURS TEST.

| NST method | # Hours (h) | CER | | | |
|---|---|---|---|---|---|
| | | GigaSpeech 2 DEV | TEST | Common Voice TEST | FLEURS TEST |
| Sys. 1 (Tab. 2, iter 2 → iter 3) | 7219 | 10.47 | 12.38 | 4.63 | 10.96 |
| Sys. 2 (Tab. 2, iter 2 → iter 3, without relabeling) | 7219 | $10.77_{+2.9\%}$ | $12.90_{+4.2\%}$ | $5.23_{+13.0\%}$ | $10.72_{-2.2\%}$ |
| Sys. 3 (Tab. 2, iter 2 → iter 3, larger augmentation) | 7219 | $10.65_{+1.7\%}$ | $12.81_{+3.5\%}$ | $5.36_{+15.8\%}$ | $10.86_{-0.9\%}$ |

