# OpenReview forum: "GigaSpeech 2: An Evolving, Large-Scale and Multi-domain ASR Corpus for Low-Resource Languages with Automated Crawling, Transcription and Refinement"
_ICLR.cc/2025/Conference — ICLR 2025 Conference Withdrawn Submission_

### Official Review · Reviewer_9q8x · 2024-10-18

**Soundness:** 2
**Presentation:** 2
**Contribution:** 2
**Rating:** 3
**Confidence:** 4

**Summary:**

This paper presents GigaSpeech 2, an evolving large-scale (30000 hours) multilingual ASR corpus for Thai, Indonesian, and Vietnamese. The dataset is constructed in the following steps: crawled video/audio from YouTube, automatically transcribed using Whisper, forced-aligned with Torchaudio, policy-based filtering, and finally label-refinement with noisy-student-training (NST). A modified NST approach is used to obtain higher-quality labels through iterative refinement, where an ASR model is trained on a subset of noisy data, generates labels for a larger subset, and that generated data is used to train a larger ASR model. Results of some representative ASR models are also reported. It is shown that the model trained with iteratively refined datasets has the SOTA performance on the test datasets.

**Strengths:**

- Dataset for speech is very important, and the speech community is thriving to see a new massive dataset.
- The proposed data creation flow is clear, and code is provided for reproducibility.
- The provided leaderboard gives the community a better sense of data difficulty. The reported WERs make me more convinced that the test set is realistic and applicable.

**Weaknesses:**

## Writing
- W1: Please use the citation format correctly. It should be \citep and \citet, depending on the usage. It seems like the current version is a direct transfer from NeurIPS, which uses \cite.
- W2: The organization of the paper is a little messy. There are too many bold texts with multiple hierarchies (for example Sec 3.1). At the same time, there are also numbered points in Sec 4. When finish reading the paper, I already get lost on what the contributions of the paper are. I suggest making the paper more organized. One simple improvement is to reference the section numbers in your contribution in the Introduction.
 - W3: The discussion of related works is a little weak. You should mention other works that also automatically create audio datasets. For example, [1],[2], [3]. There should be at least one subsection in related works that discusses this topic.

## Novelty
 - W4: While I appreciate the release of a new dataset, I don't find the dataset curation approach much novel. In particular, everything besides the NST seems very trivial and purely engineering work. The proposed NST is similar to the approach taken by [3]. I would like the authors to provide a detailed explanation of how their approach differs from previous approaches.

## Evaluation
 - W5: $\textbf{This is the biggest issue: }$ How good exactly is the training dataset? There are only evaluation results of fine-tuned models on the test dataset, but to claim a usable dataset, you $\textbf{must}$ report the accuracy of the dataset (at least on a random subset). Currently, such disclosure on the training set is missing.
- W6: There have been concerns raised by previous works on training with synthetic datasets. For example, I would like to quote Whisper's paper [4]: "Recent research has shown that training on datasets of mixed human and machine-generated data can significantly impair the performance of translation systems (Ghorbani et al., 2021)." I would like the authors to provide a more detailed analysis of whether their NST approach would solve that problem. It will also be a significant finding if self-training will completely solve that problem, which means synthetical datasets could be used in the future.
- W7: If I understand correctly, Table 3 is comparing pre-trained models with fine-tuned GigaSpeech2 model. This seems a little unfair. If possible, I would like to see fine-tuned public models as well.

## Reference

[1] Sun, Luoyi, et al. "Auto-ACD: A large-scale dataset for audio-language representation learning." ACM Multimedia 2024. 2024.

[2] Liu, Houjun, et al. "Automation of language sample analysis." Journal of Speech, Language, and Hearing Research 66.7 (2023): 2421-2433.

[3] Yeroyan, Ara, and Nikolay Karpov. "Enabling ASR for Low-Resource Languages: A Comprehensive Dataset Creation Approach." arXiv preprint arXiv:2406.01446 (2024).

[4] Radford, Alec, et al. "Robust speech recognition via large-scale weak supervision." International conference on machine learning. PMLR, 2023.

**Questions:**

Please answer the weaknesses. Based on your resources, I recommend answering the weakness in the following order: W5, W4, W6, W7, W3, W2, W1. I will reconsider the score if the authors promptly address W5 and W4.

---

### Official Review · Reviewer_Ldsz · 2024-10-23

**Soundness:** 3
**Presentation:** 4
**Contribution:** 3
**Rating:** 6
**Confidence:** 5

**Summary:**

Authors introduce a dataset (Gigaspeech 2) made up of 30,000h of multilingual speech for low resource languages (such as Thai, Indonesian, Vietnamese) crawled on youtube and automatically transcribed with Whisper large model followed by data quality filtering iterative process.
A modified version of the Noisy Student Training (NST) method by Xie et al. (2020) is notably proposed to iteratively refine labels from collected (noisy) data.
ASR models trained using this corpus improve upon Whisper 3 large baseline (and other commercial systems) for Thai and have similar results for Vietnamese and Indonesian, demonstrating quality of dataset (for languages such as Thai, Indonesian, Vietnamese).

**Strengths:**

-strong ASR resource for 3 languages: indo, vn and thai => strong impact for those 3 languages

-methodology (multiple iteration approach) is applicable to many other languages

**Weaknesses:**

-this is overall a good recipe for collecting large and good quality speech resources, but it applies well-known techniques and eventually we get a resource with 3 languages covered only (extending language coverage is mentioned as future work)

-for readers not familar with NST method, section 3.2 is not easy to understand and would benefit from a bit more context on base NFT method before describing how it was modified by the authors

-the comparison to existing ASR systems is convincing for Thai but less convincing for Vietnamese and Indonesian (although it can be that commercial systems that outperform the Gigaspeech 2 based ASR system might have been trained on commonvoice and fleurs datasets)

-multidimensional filtering and its impact could have been more detailed (how much data is rejected for each specific filter, etc.)

**Questions:**

-« To ensure no speaker overlap between the splits, we manually verify no speaker overlap between different channels and partition the data by allocating different YouTube channels to each subset. » => How can you be sure ?
Same speaker might speak in multiple channels (especially celebrities). Do you handle this specifically ?

-Corpus made available to the community ? Which license do you plan to have ?

-What is the impact of the seed (1st iteration) ASR system’s (Whipser3's) performance in your case ? I mean, it probably has different performance on the 3 languages targeted, how does this impact the final quality of the dataset and final ASR performance for a given language ?

-text normalization: i know the practice in ASR is to remove punct and case but since Whisper 3 is providing it, why not keeping it as a second tier of labels ? COuld be interesting for NLP researchers who need case/punct !

-« Moreover, the collected audio must be accompanied by a Creative Commons license. »  but which one, there are several CC licenses with different rules !

---

### Official Review · Reviewer_5Eop · 2024-11-02

**Soundness:** 3
**Presentation:** 3
**Contribution:** 2
**Rating:** 6
**Confidence:** 5

**Summary:**

This paper presents GigaSpeech 2, a large-scale, multi-domain, and multilingual ASR corpus targeting low-resource Southeast Asian languages, specifically Thai, Indonesian, and Vietnamese. It introduces a data collection, and refinement pipeline focused on automated transcription and label refinement using Noisy Student Training (NST), aiming to improve ASR systems for low-resource languages with minimal reliance on labelled data.

**Strengths:**

- Gigaspeech 2 test sets: a domain-rich human labelled benchmark enabling realistic ASR performance evaluation for Thai, Indonesian, and Vietnamese.
- A comprehensive speech dataset featuring 30K hours of raw audio across three low-resource Southeast Asian languages: Thai, Indonesian, and Vietnamese.
- High-performance ASR systems for the three languages, on par with commercially available solutions.

**Weaknesses:**

- Although GigaSpeech 2 spans multiple domains, the performance drops on out-of-domain test sets (Common Voice, FLEURS) in Indonesian and Vietnamese suggest limitations in cross-domain generalizability (Table 3).
- Even though the authors mention zero reliance on the labelled data, the approach is strictly dependent on having an initial good seed model which can be used for transcription and timestamp prediction; hence, this approach does not completely preclude the need for labelled data.
- The quality of the training data released as a part of this work depends on the robustness of the Whisper model

**Questions:**

- In Table 2, it is not clear if the improvements in WER are coming because of the increased number of refined hours or due to the added model parameters (it would have been best to keep the model parameters the same and just make use of additional refined data).
- In Table 4, can the authors please share some insights on the disparities between the performance of the 68.6M model and the 151.9M model? In some cases, the larger model is performing better, and the differences are quite significant (which ideally should not be the case).

---

### Official Review · Reviewer_By99 · 2024-11-02

**Soundness:** 4
**Presentation:** 4
**Contribution:** 2
**Rating:** 3
**Confidence:** 4

**Summary:**

This work presents Gigaspeech2, which contains 30k hours of automatically transcribed speech of Thai, Indonesian, and Vietnamese.

Authors introduce a pipeline to download and process the dataset by

- collect dataset from Youtube with proper filtering
- force align with torchaudio
- transcribe with Whisper
- iteratively improve the transcription quality by noisy student training

Authors demonstrate through a few experiments that it achieved great ASR performance on target languages (i.e. Thai, Indonesian and Vietnamese) when compared with other commercial models and open-source models

**Strengths:**

The speech dataset from Thai, Indonesian, and Vietnamese are usually limited, this Gigaspeech2 would be beneficial to the speech community which target those languages.

While the target languages is limited, the experiments comparing with other commercial models, open-source models and datasets are convincing. They demonstrate the proposed pipeline and collected dataset achieves good quality (for example compared with YODAS dataset)

**Weaknesses:**

Despite the usefulness of the dataset itself, the main weakness is that the general novelty is limited in this work. The pipeline itself does not have many components that are significantly different from the existing dataset collection procedures. For example, one of the main contribution authors claim is to apply Noisy Student Training iteratively to refine collected dataset, however, this is a quite standard approach and is already applied in one of the original Noisy student training paper (e.g. https://arxiv.org/pdf/1911.04252). It would be great that authors can provide more novelty how their refinement can be distinguished from the existing approach.

Another minor concern is this dataset only contains Thai, Indonesian, and Vietnamese, and their target audience might be too limited in the context of ICLR. It would be nice if authors can extend their approach to much more languages and demonstrate they are useful.

**Questions:**

The iteration seems to still have a lot of room even after 2,3 iterations, how much would authors expect the result to improve through a few more iterations?

---

### Official Review · Reviewer_Ah5s · 2024-11-10

**Soundness:** 3
**Presentation:** 3
**Contribution:** 2
**Rating:** 1
**Confidence:** 5

**Summary:**

The paper presents a 30k hr multidomain, multilingual ASR corpus for low-resource Southeast Asian languages, termed GigaSpeech 2.
It describes the assembly of the data set - data crawling, transcription with Whisper ASR, forced alignment with TorchAudio, text normalization, and a modified Noisy Student Training scheme to improve the labels. It presents results on test sets from Thai, Indonesian, and Vietnamese compared to open-source and commercial models.

**Strengths:**

The language resource  will be of value to the ASR community. The methodology for constructing the data set is technically sound, though more advanced methods for e.g. forced alignment exist (only off-the-shelf tools were used here). Results demonstrate the value of the training data.

**Weaknesses:**

The paper describes construction of a language resource and empirical ASR results demonstrating essentially that training on more relevant data helps lower WERs. Evaluation was on 3 languages only. Most importantly, the paper does not present a scientific hypothesis or any advancement in methodologies; it is purely a resource and evaluation paper and as such would be better place in a speech conference or language resources conference.

**Questions:**

For a revised submission it would be nice to see results on more languages .

---

### Note · Authors · 2024-11-21

**Comment:**

It's a shame that the ICLR community has mean reviewer like Ah5s. The review was **excessively brief and subjective**, offering a **very strong reject (1) with absolute confidence (5)** but without providing the necessary depth or constructive feedback.

**Withdrawal Confirmation:**

I have read and agree with the venue's withdrawal policy on behalf of myself and my co-authors.